



# A cluster-of-functional-groups approach for studying organic enhanced atmospheric cluster formation

Astrid Nørskov Pedersen, Yosef Knattrup, and Jonas Elm

Department of Chemistry, Aarhus University, Langelandsgade 140, 8000 Aarhus C, Denmark

**Correspondence:** Jonas Elm (jelm@chem.au.dk)

**Abstract.** The role of organic compounds in atmospheric new particle formation is difficult to disentangle due to the myriad of potentially important oxygenated organic molecules (OOMs) present in the atmosphere. Using state-of-the-art quantum chemical methods, we here employ a novel approach, denoted the "cluster-of-functional-groups" approach, for studying the involvement of OOMs in atmospheric cluster formation. Instead of the usual "trial-and-error" approach of testing the ability of

experimentally identified OOMs to form stable clusters with other nucleation precursors, we here study which, and how many, intermolecular interactions that are required in a given OOM to form stable clusters. In this manner we can reverse engineer the elusive structure of OOM candidates that might be involved in organic enhanced atmospheric cluster formation.

We calculated the binding free energies of all combinations of donor/acceptor organic functional groups to investigate which functional groups that most preferentially bind with each other and with other nucleation precursors such as sulfuric acid

and bases (ammonia, methyl-, dimethyl-, and trimethylamine). We find that multiple carboxyl groups leads to substantially more stable clusters compared to all other combinations of functional groups. Employing cluster dynamics simulations, we investigate how a hypothetically OOM composed of multiple carboxyl groups can stablize sulfuric acid – base clusters and provide recommendations for potential atmospheric multi-carboxylic acid tracer compounds that should be explicitly studied in the future.

The presented "cluster-of-functional-groups" approach is generally applicable and can be employed in many other applications, such as ion-induced nucleation and potentially in elucidating the structural patterns in molecules that facilitate ice nucleation.

## 1  Introduction

Secondary organic aerosols (SOA) constitute a major fraction of organic matter in the atmosphere (Jimenez et al., 2009)

and influence both human health (Pelucchi et al., 2009) and global climate (Canadell et al., 2021). The term SOA formation usually comprises both the initial aerosol nucleation and the subsequent growth of existing aerosol particles via vapour uptake. However, while oxygentaed organic molecules (OOMs) are widely accepted as being important for aerosol growth, it remain ambiguous whether OOMs are important for aerosol nucleation in the atmosphere (Kirkby et al., 2023).

The onset of aerosol nucleation is governed by the formation of stable atmospheric molecular clusters (Kulmala et al., 2013).

Sulfuric acid (Sipilä et al., 2010) and atmospheric bases such as ammonia (AM) (Kirkby et al., 2011) and amines (Almeida





et al., 2013; Jen et al., 2014; Glasoe et al., 2015) (methylamine (MA), dimethylamine (DMA), and trimetheylamine (TMA)) are important species for nucleation both over land and sea. In addition, iodine species have been shown to contribute to nucleation at coastal and marine environments (Baccarini et al., 2020; Beck et al., 2021; He et al., 2021, 2023). The global modelling study by Dunne et el. (Dunne et al., 2016) indicated that almost all nucleation occurring in the atmosphere involves sulfuric

acid coupled with either ammonia and/or organics. However, the exact role of organics in aerosol nucleation remains elusive. Pure organic ion-induced nucleation has been reported both at the CLOUD chamber (Kirkby et al., 2016) and in the field (Rose et al., 2018). While some early work has inferred that organics play a role in multicomponent cluster formation (Zhang et al., 2004; Metzger et al., 2010; Schobesberger et al., 2013; Riccobono et al., 2014), it remains unknown whether this is also true for multicomponent SA–base–OOM nucleation or whether it proceeds via two isolated pathways (one for SA–base and one for

pure OOMs). Both experiments (Lehtipalo et al., 2018; Kirkby et al., 2023) and quantum chemical calculations (Elm, 2019b) allude to a decoupled mechanism, but this has not yet been definitively confirmed.

The puzzle of the role of organics in aerosol nucleation originate from the fact that there exist a myriad of OOMs in the atmosphere. Volatile organic compounds (VOCs) are rapidly transformed into less volatile species in the air due to oxidation reactions with either OH, $NO_3$ or $O_3$, making it challenging to disentangle which OOMs are important for nucleation and

which are important for SOA growth. Identification of OOMs is usually performed with mass spectrometer techniques such as the CI-APi-TOF (Jokinen et al., 2012). For instance, the seminal work by Ehn et al. (Ehn et al., 2014) measured the chemical composition of OOM monomers ($C_{10}H_{14-16}O_{7-11}$) and OOM covalently bound dimers ($C_{19-20}H_{28-32}O_{10-18}$) from $\alpha$-pinene ($C_{10}H_{16}$) oxidation. While this gives important information about the chemical composition of the OOMs, it yields no information about the exact functional groups or their arrangement in the molecules. Quantum chemical (QC) calculations can be

applied to yield the cluster structures and in conjugation with cluster dynamics modelling can elucidate the role of specific OOMs in nucleation. However, this requires that the given OOM structure is known. This has led to an overabundance of QC studies, including work by our group, that investigate the binding affinity of different experimentally identified OOMs to themselves and other nucleation precursors. We recently reviewed the entire QC literature on the role of organics in cluster formation and unfortunately, not a single OOM has definitively been proven to participate in nucleation in the planetary

boundary layer (Elm et al., 2023). The lack of progress could be ascribed to the fact that previous work have been looking at the wrong compounds. All studies have been performed on the organic monomers, while recent evidence from the CLOUD chamber has shown that it is in fact the covalently bound organic dimers, which have ultra low volatilities (ULVOCs) (Simon et al., 2020) that drive nucleation (Lehtipalo et al., 2018; Dada et al., 2023). We will hereon refer to these dimers as accretion products to clearly distinct them from dimer clusters. Studying large $C_{19-20}H_{28-32}O_{10-18}$ OOM accretion products such as those

from $\alpha$-pinene oxidation is challenging using QC methods and to date not a single study exists. This is caused by the fact that QC methods scale steeply with system size. In addition, larger, more flexible molecules can exist in numerous conformations, which rapidly increases number of calculations required to identify the global minimum cluster structures.

Using state-of-the-art quantum chemical methods, we here employ a new approach, denoted the "cluster-of-functional-groups" approach (Elm et al., 2023), for studying the involvement of OOMs in atmospheric cluster formation. Instead of the

usual "trial-and-error" approach of testing the ability of identified OOMs to form stable clusters with themselves or other nu-



cleation precursors, we here study exactly which and how many intermolecular interactions (in the form of functional groups) that are required in a given OOM to form stable clusters. This allows us to reverse engineer the potential structure of OOMs involved in organic enhanced atmospheric cluster formation. We explicitly study all possible donors (alcohol, peroxide) and acceptors (ether, epoxide, aldehyde, ketone, acid anhydride and ester), as well as carboxylic acids. Based on the functional groups that bind strongest, we extend the analysis to study the cluster dynamics of a hypothetical OOM binding to $(SA)_{1-2}(Base)_{1-2}$ clusters, where the bases are AM, MA, DMA and TMA.

## 2 Methods

### 2.1 Computational details

The semi-empirical GFN1-xTB (Grimme et al., 2017) energy calculations and geometry optimization were calculated with the xtb 6.4.0 program (Bannwarth et al., 2021). Gaussian16, version B.01 (Gau) was used for the Density Functional Theory (DFT) calculations with default convergence criteria. The $\omega$B97X-D functional (Chai and Martin, 2008) with the 6-31++G(d,p) basis set was chosen for geometry optimization and vibrational frequency calculations based on its performance in numerous benchmarks (Elm and Mikkelsen, 2014; Myllys et al., 2016; Elm and Kristensen, 2017; Schmitz and Elm, 2020; Jensen et al., 2022). Given that some vibrational frequencies were low, the quasi-harmonic approximation, as recommended by Grimme, (Grimme, 2012) was used to treat vibrational frequencies below $100 \text{ cm}^{-1}$ as free rotors. ORCA 5.0.4 (Neese, 2012, 2018; Neese et al., 2020; Neese, 2022) was used to calculate single-point energy corrections using the Domain Local Pair Natural Orbital, DLPNO–CCSD(T$_0$) (Riplinger and Neese, 2013; Riplinger et al., 2013) with the aug-cc-pVTZ basis set, using the TightSCF convergence criteria and the NormalPNO setting (Liakos et al., 2015). The workflow and subsequent data processing was automated using the JKCS program (Kubečka et al., 2023). All data have been added to the Atmospheric Cluster DataBase (ACDB) (Elm, 2019a). The atmospheric cluster dynamics were simulated using the Atmospheric Cluster Dynamics Code (ACDC) (McGrath et al., 2012) using a modified version of the code given in the online repository by Tinja Olenius (Olenius et al., 2013; Roldin et al., 2019) available in the JKCS suite. The simulations were done at 278.15 K with a constant coagulation sink of $-1.6 \times 10^{-3} \text{ s}^{-1}$, matching typical sink values (Dal Maso et al., 2008; Kontkanen et al., 2017). The clusters with a single sulfuric acid monomer were counted towards the total sulfuric acid concentration.

### 2.2 Configurational sampling

For the configurational sampling procedure, a funnel-type approach was employed (Temelso et al., 2018; Odbadrakh et al., 2020; Kubečka et al., 2019), where the level of theory is increased in each step as the number of candidate cluster structures is decreased. The workflow can be described as follows:

ABCluster → GFN1-xTB$^{\text{OPT}}$ → ArbAlign → $\omega$B97X-D$^{\text{OPT+FREQ}}$ → DLPNO$^{\text{SP}}$

ABCluster (Zhang and Dolg, 2015, 2016) was used to generate a large pool of cluster structures with the CHARMM forcefield. Settings as recommended by Kubečka et al. (Kubečka et al., 2019) with population size, $SN = 3000$, maximum generations,

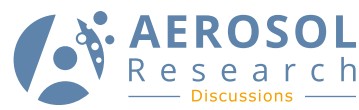

$g_{max} = 200$, and number of scout bees, $g_{limit} = 4$ was used. For each protonation state in the cluster, 1000 local minima were saved. All local minima were optimized at the GFN1-xTB level. ArbAlign (Temelso et al., 2017) was used to filter out identical clusters, based on root-mean-square deviation (RMSD) values. Based on previous studies on atmospheric clusters, (Kildgaard

et al., 2018b, a) an RMSD cutoff of 0.38 Å was chosen. As a large number of structures remained after ArbAlign, only the 100 cluster structures with the lowest electronic energy was selected for further optimization at the $\omega$B97X-D/6-31++G(d,p) level of theory. The five structures with the lowest free energy were then selected for calculation of single-point energies at the DLPNO-CCSD(T$_0$)/aug-cc-pVTZ-level.

### 2.3  The "cluster-of-functional-groups" approach

Small organic molecules were chosen to represent the functional groups that act as hydrogen bond donors (alcohol (CH$_3$OH) and peroxide (CH$_3$OOH)), as well as functional groups that act as hydrogen bond acceptors (ether (CH$_3$OCH$_3$), epoxide (C$_2$H$_4$O), aldehyde (CH$_3$CHO), ketone (CH$_3$COCH$_3$), acid anhydride (CH$_3$C(=O)OC(=O)CH$_3$) and ester (COOCH$_3$)). Besides these groups, carboxylic acid, (HCOOH), which is both an acceptor and a donor, was also included. The studied functional groups were included in the funnelling approach in order to identify the global minimum cluster structures. In this manner the

functional groups should orient in the most favourable positions and can be viewed as ideal "contact points" for assembling the molecular backbone afterwards. Hence, the "cluster-of-functional-groups" should be viewed as a single proxy OOM. The strength of this approach is that we do not need to explicitly consider the origin of the OOM i.e anthropogenic (aromatics) or biogenic (isoprene/terpenes) as the approach will inherently identify the structural patterns that are important directly based on their ability to participate in cluster formation (Elm et al., 2023). However, we note the caveat that the "cluster-of-functional-

groups" approach assumes that the binding free energies of the individual groups are additive. This might not necessarily be the case for realistic atmospheric OOMs, but the approach can still be employed for screening purposes and to yield some indication of which combinations of functional group might potentially be important in atmospheric cluster formation.

### 2.4  Atmospheric cluster dynamics

To study the time development of molecular cluster distributions, the free energy is used to solve the birth-death equations with

ACDC (McGrath et al., 2012). The birth-death equations describe how clusters are destroyed and created by condensation and evaporation and are given as the change in concentration of cluster $i$,

$$\frac{dc_i}{dt} = \sum_{j<i} \beta_{j,(i-j)} c_j c_{(i-j)} + \sum_j \gamma_{(i+j)\rightarrow i} c_{(i+j)} - \sum_j \beta_{i,j} c_i c_j - \sum_{j<i} \gamma_{i\rightarrow j} c_i + Q_i - S_i. \tag{1}$$

Here, $j$ is another cluster in the system, $\beta_{i,j}$ is the collision coefficient between cluster $i$ and $j$, $\gamma_{i\rightarrow j}$ is the evaporation coefficient of cluster $i$ into a smaller cluster, of which one is cluster $j$. $Q_i$ covers outside sources of $i$, and $S_i$ other possible

loss mechanisms for $i$. The collision coefficient is calculated using kinetic gas theory and the evaporation coefficient via mass balance based on calculated free energy,

$$\gamma_{(i+j)\rightarrow i} = \beta_{i,j} c_{ref} \exp\left(\frac{\Delta G_{i+j} - \Delta G_i - \Delta G_j}{k_b T}\right), \tag{2}$$





where $c_{\text{ref}}$ is the monomer concentration of the vapour used to calculate the free energies. The birth-death equations are generated by checking all possible cluster configurations and examining which evaporations and collisions are able to destroy or

create a given cluster (McGrath et al., 2012). As it is not possible to explicitly simulate the entire cluster size range, a user-specific size limit has to be chosen where the clusters are considered stable against evaporation and are counted towards the formation rate. We will refer to these clusters as "outgrowing". For our systems, we chose the clusters with one additional acid compared to the cluster sizes we have data for [$(SA)_3(base)_2$, $(SA)_2(OOM)_2(base)_2$ and $(SA)_3(OOM)_1(base)_2$]. These outgrowing cluster sizes are quite small and therefore artificially stabilize the systems as the critical cluster size is not nec-

essarily captured well. To distinguish the calculated rates from actual nucleation rates, we will refer to the rates as "cluster formation potentials" ($J_{\text{potential}}$). These rates illustrates the potential of the cluster to grow to larger sizes and corresponds to an upper-bound on the formation rate. We refer to the clusteromics I paper (Elm, 2021a) for additional information on "cluster formation potentials".

## 3   Results and discussion

### 3.1   Pure organic clusters

Initially, pure organic clusters were studied containing every possible donor-acceptor combination. This lead to a total of 21 dimer cluster structures. The calculated free energies, at the DLPNO-CCSD(T$_0$)/aug-cc-pVTZ//$\omega$B97X-D/6-31++G(d,p) level of theory, are given in Figure 1. The calculations are performed at 298.15 K and 1 atm.



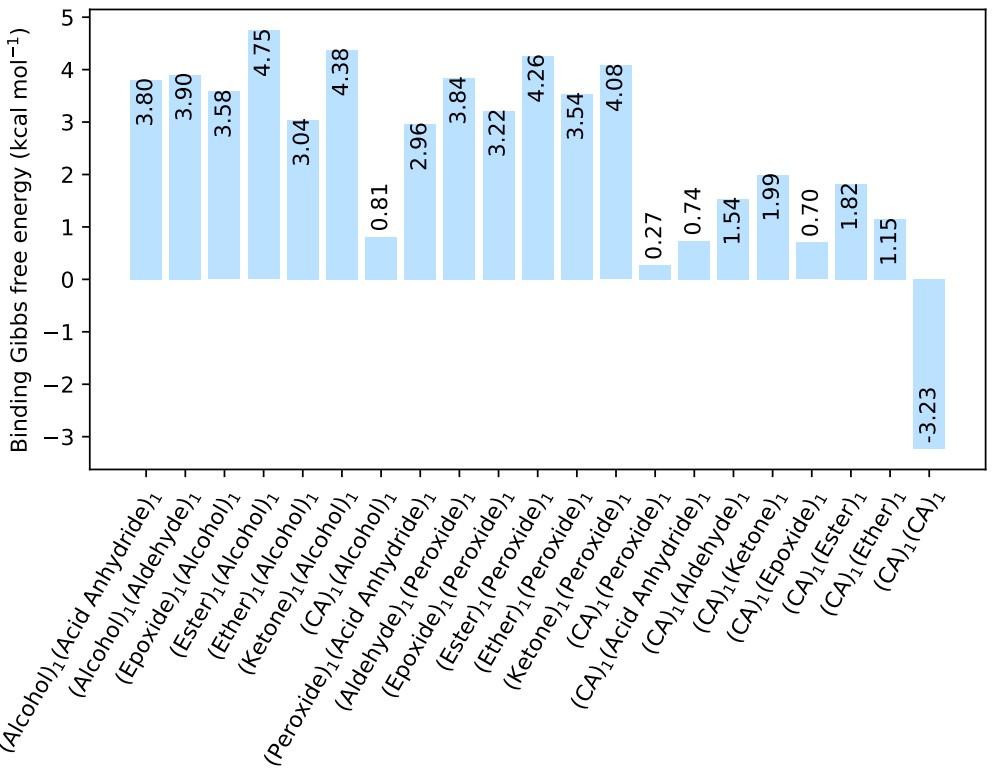

**Figure 1.** The binding free energy of all organic-organic interactions between all combinations of hydrogen bond donor and acceptor, calculated at the DLPNO-CCSD($T_0$)/aug-cc-pVTZ //$\omega$B97X-D/6-31++G(d,p) level of theory with quasi-harmonic cutoff of 100 cm$^{-1}$, at 298.15 K and 1 atm. CA is short for Carboxylic Acid.

All the dimer clusters have a positive change in free energy, except for the dimer cluster with two carboxylic acid (CA) groups that has a $\Delta G = -3.23$ kcal mol$^{-1}$. This value is in excellent agreement with the experimental value of $-3.46$ kcal mol$^{-1}$ for the formic acid dimer cluster given by Vander et al. (Vander Auwera et al., 2007), at 296 K. However, this value correspond to a high evaporation rate of the formic acid dimer. Thus, cluster formation involving only organic molecules is unlikely at 298.15 K and 1 atm, unless the molecule contains multiple carboxylic acid groups. In all cases the inclusion of a carboxyl group lowers

the binding free energy considerably compared to the other functional groups. Hence, it is likely that other compounds besides organics are needed for forming the initial stable clusters.

## 3.2   SA–Organic clusters

As sulfuric acid is known to be essential for new particle formation, dimer clusters involving all the different organic functional groups and SA is studied. We studied the $(SA)_1(organic)_1$ dimer clusters where only a single organic functional group is

present, as well as the $(SA)_1(CA)_1(organic)_1$ clusters that have been fully saturated by organic functional groups, meaning all donor and acceptor groups in the SA molecule are hydrogen-bonded. This led to a total of 46 clusters studied. The fully

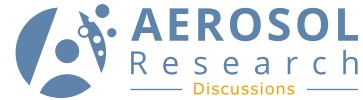

saturated SA clusters all include at least one carboxylic acid group as well as two other identical organic groups, except for the case where there is two carboxylic acid groups in total as this fully saturates the SA molecule. The calculated free energies (at 298.15 K and 1 atm) are shown in Figure 2.

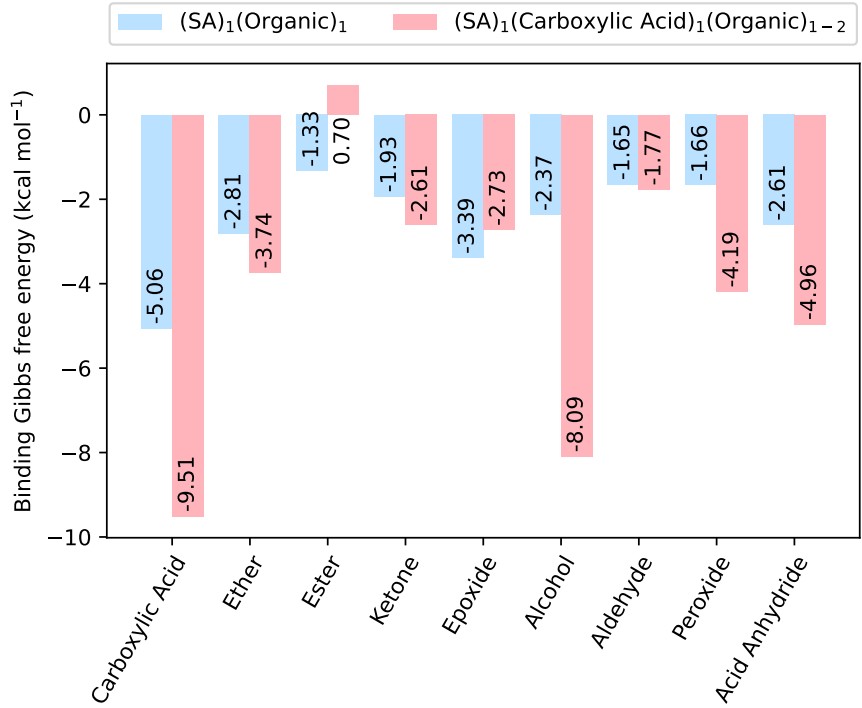

**Figure 2.** The binding free energy of clusters with one SA and one organic functional group, as well as with one SA fully saturated by organic functional groups, calculated at the DLPNO-CCSD($T_0$)/aug-cc-pVTZ//$\omega$B97X-D/6-31++G(d,p) level of theory with quasi-harmonic cutoff of 100 cm$^{-1}$, at 298.15 K and 1 atm.

As expected, including SA decreases the free energy for all the dimer clusters when compared to the pure organic dimer clusters. Generally, saturating the SA molecule with organic functional groups also results in a cluster lower in free energy. However, in cases where there is added two hydrogen bond donors to the $(SA)_1(CA)_1$ cluster, the addition of the second functional group leads to an increase in the binding free energy. For instance, this is the case for the clusters containing either two ester and two epoxide.

For the dimer clusters, $(SA)_1(CA)_1$ is the most stable with $\Delta G = -5.06$ kcal mol$^{-1}$, while for the clusters with a fully saturated SA, it is $(SA)_1(CA)_2$, with $\Delta G = -9.51$ kcal mol$^{-1}$, indicating a trend of carboxylic acid being the most stabilizing functional group. This finding is consistent with the previous study by Elm et al. (Elm et al., 2017a). It should be noted that the $(SA)_1(CA)_1(Alcohol)_2$ cluster has a binding free energy very close to that of the $(SA)_1(CA)_2$ cluster, with a value of $\Delta G = -8.09$ kcal mol$^{-1}$. This suggests that multiple alcohol groups could also be important for cluster formation.





### 3.3 (SA)$_1$(base)$_1$–organic clusters

Next, bases are added to the clusters, to gauge its effect on the cluster stability. Initially, simple trimer clusters involving one organic functional group, one SA, and one base (either AM, MA, DMA or TMA) are studied. The calculated binding free energies of these clusters, at 298.15 K and 1 atm are presented in Figure 3.

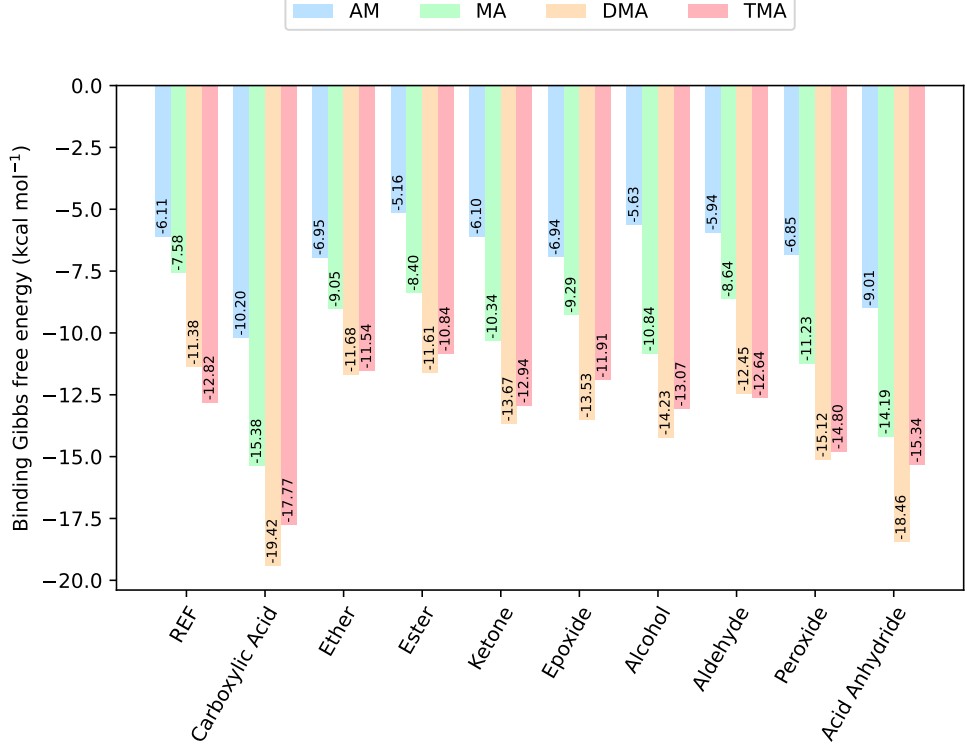


**Figure 3.** The binding free energy of the (SA)$_1$(Base)$_1$(Organic)$_1$ clusters, calculated at the DLPNO-CCSD(T$_0$)/aug-cc-pVTZ//$\omega$B97X-D/6-31++G(d,p) level of theory with quasi-harmonic cutoff of 100 cm$^{-1}$, at 298.15 K and 1 atm. Carboxylic acid, the base, and sulfuric acid has been omitted in the labels for readability. REF is the reference (SA)$_1$(Base)$_1$ cluster without organics present.

In all cases the (SA)$_1$(base)$_1$(organic)$_1$ clusters are lower in free energy compared to the (SA)$_1$(base)$_1$ clusters. Hence, the addition of organics is thermodynamically favourable. For instance, the addition of a carboxylic acid group to the (SA)$_1$(base)$_1$ leads to a lowering of the free energy by $-4.09$, $-7.80$, $-8.09$ and $-4.95$ kcal mol$^{-1}$ for AM, MA, DMA and TMA, respectively.

Generally, the cluster binding free energies follow the basicity of the base with the following trend: DMA $\simeq$ TMA $<$ MA $<$ AM. This is consistent with previous work, where it has been shown both theoretically (Kurtén et al., 2008) and experimentally (Jen et al., 2014; Glasoe et al., 2015; Almeida et al., 2013) that amines are much more effective in driving NPF with SA than AM is, even when accounting for the large difference in atmospheric mixing ratios. The ordering of DMA and TMA depends




slightly on the system, with DMA dominantly leading to the most stable clusters. In particular, the $(SA)_1(DMA)_1(CA)_1$,

$(SA)_1(TMA)_1(CA)_1$ and $(SA)_1(DMA)_1(Acid\ Anhydride)_1$ clusters have the lowest free energies, with $\Delta G$ of -19.42, -17.77 and -18.46 kcal mol$^{-1}$, respectively. Carboxylic acid groups are again found to form the most favourable interactions, with the cluster with acid anhydride only being about 1 kcal mol$^{-1}$ less stable.

To further explore which functional groups that could yield stable clusters, we added one more organic functional group to the $(SA)_1(Base)_1$ clusters. As carboxylic acid has been prevalent in forming the most stable clusters in the previous sections we

added the constraint that one of the organics should be a carboxylic acid forming the $(SA)_1(Base)_1(CA)_1(Organic)_1$ clusters. The calculated binding free energies (at 298.15 K and 1 atm) are given in Figure 4.

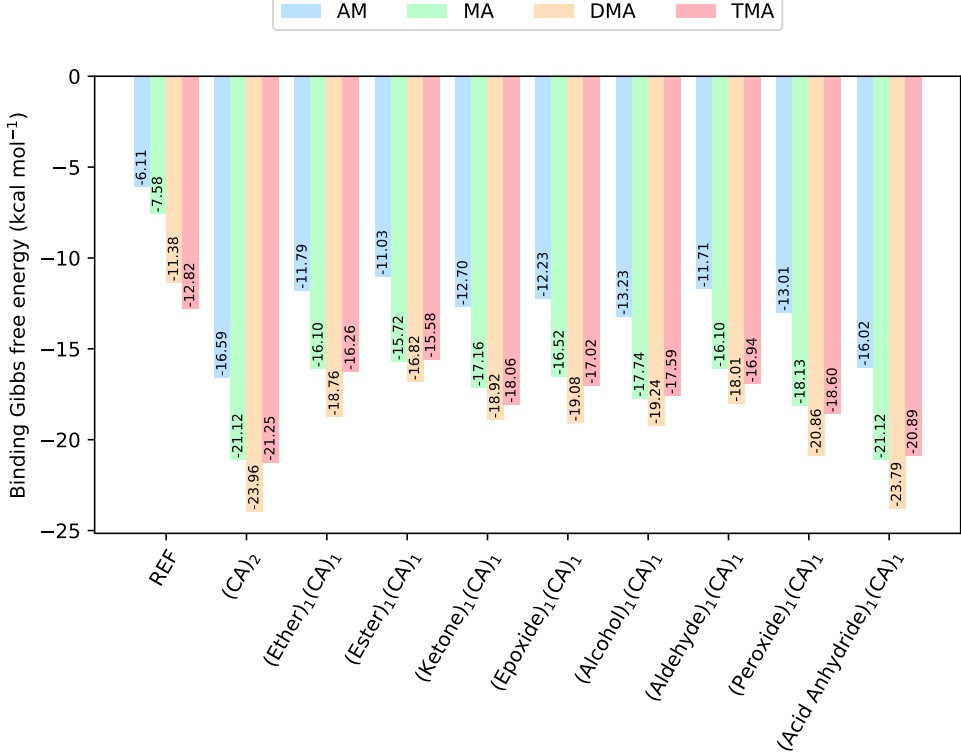

**Figure 4.** The binding free energy of the $(SA)_1(Base)_1(CA)_1(Organic)_1$ clusters, calculated at the DLPNO-CCSD(T$_0$)/aug-cc-pVTZ//$\omega$B97X-D/6-31++G(d,p) level of theory with quasi-harmonic cutoff of 100 cm$^{-1}$, at 298.15 K and 1 atm. CA is short for Carboxylic Acid. The base and sulfuric acid has been omitted in the labels for readability. REF is the reference $(SA)_1(Base)_1$ cluster without organics present.

The addition of one more organic functional group has a stabilizing effect on all the clusters. The cluster with two carboxylic acids is found to be lowest in free energy, with $\Delta G = -23.96$ kcal mol$^{-1}$ for the $(SA)_1(DMA)_1(CA)_2$ cluster. However, the

cluster that contains an acid anhydride is also very close in free energy, with $\Delta G = -23.79$ kcal mol$^{-1}$. The cluster structures




can be seen in Figure 5, calculated at the DLPNO-CCSD($T_0$)/aug-cc-pVTZ//$\omega$B97X-D/6-31++G(d,p) level of theory. In both cases the organics can be seen to interact both with SA and the DMA molecules.

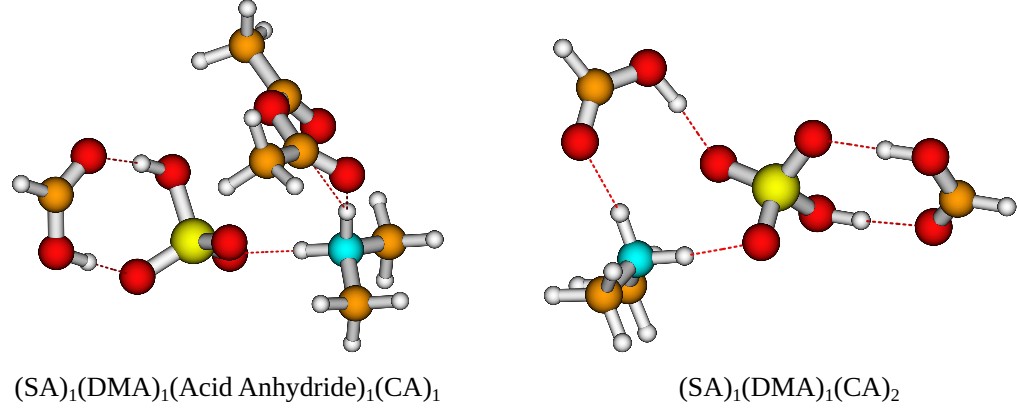

$(SA)_1(DMA)_1(Acid\ Anhydride)_1(CA)_1$ $(SA)_1(DMA)_1(CA)_2$

**Figure 5.** The $(SA)_1(DMA)_1(Acid\ Anhydride)_1(CA)_1$ and $(SA)_1(DMA)_1(CA)_2$ cluster geometries lowest in free energy. Calculated at the DLPNO-CCSD($T_0$)/aug-cc-pVTZ//$\omega$B97X-D/6-31++G(d,p) level of theory with quasi-harmonic cutoff of 100 cm$^{-1}$, at 298.15 K and 1 atm. White = hydrogen, brown = carbon, red = oxygen, yellow = sulfur, blue = nitrogen. CA is short for Carboxylic Acid.

Furthermore, the $(SA)_1(DMA)_1(Alcohol)_1(CA)_1$, $(SA)_1(DMA)_1(Aldehyde)_1(CA)_1$, and $(SA)_1(DMA)_1(Peroxide)_1(CA)_1$

clusters also have low free energy values with $\Delta G$ of $-19.24$, $-18.01$ and $-20.86$ kcal mol$^{-1}$, respectively. Figure 6 presents the cluster structures. Again the organics are interacting with both SA and the DMA. Interestingly, the aldehyde in the $(SA)_1(DMA)_1(Aldehyde)_1(CA)_1$ cluster preferentially binds to the SA and DMA compounds via several weak interactions, instead of binding to the vacant S-OH group in SA. We found that the cluster where the aldehyde reside at the S-OH group in SA is $0.62$ kcal mol$^{-1}$ higher in free than the presented structure in Figure 5 (See SI). Thus, the ability to form clusters

with organic molecules including multiple alcohols, aldehydes or peroxides as functional groups should be further studied in the future.

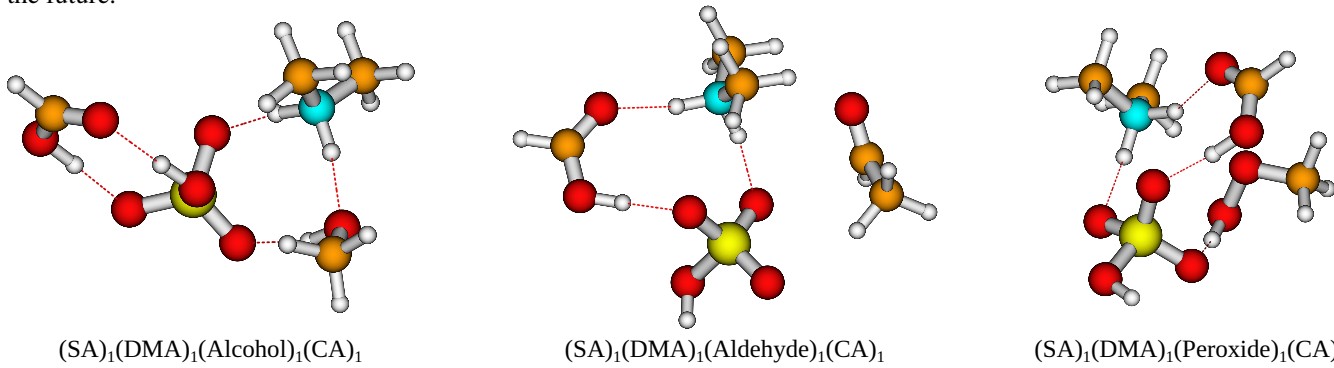

$(SA)_1(DMA)_1(Alcohol)_1(CA)_1$ $(SA)_1(DMA)_1(Aldehyde)_1(CA)_1$ $(SA)_1(DMA)_1(Peroxide)_1(CA)_1$

**Figure 6.** The $(SA)_1(DMA)_1(Alchohol/Aldehyde/Peroxide)_1(CA)_1$ cluster geometries lowest in free energy. Calculated at the DLPNO-CCSD($T_0$)/aug-cc-pVTZ//$\omega$B97X-D/6-31++G(d,p) level of theory with quasi-harmonic cutoff of 100 cm$^{-1}$, at 298.15 K and 1 atm. White = hydrogen, brown = carbon, red = oxygen, yellow = sulfur, blue = nitrogen. CA is short for Carboxylic Acid.



### 3.4 $(SA)_{1-2}(base)_{1-2}(CA)_3$ clusters

As the carboxylic acid groups consistently lead to the most stable clusters, the $(SA)_{1-2}(Base)_{1-2}(CA)_3$ clusters have also been
studied. The calculated binding free energies, at 298.15 K and 1 atm are given in Figure 7.

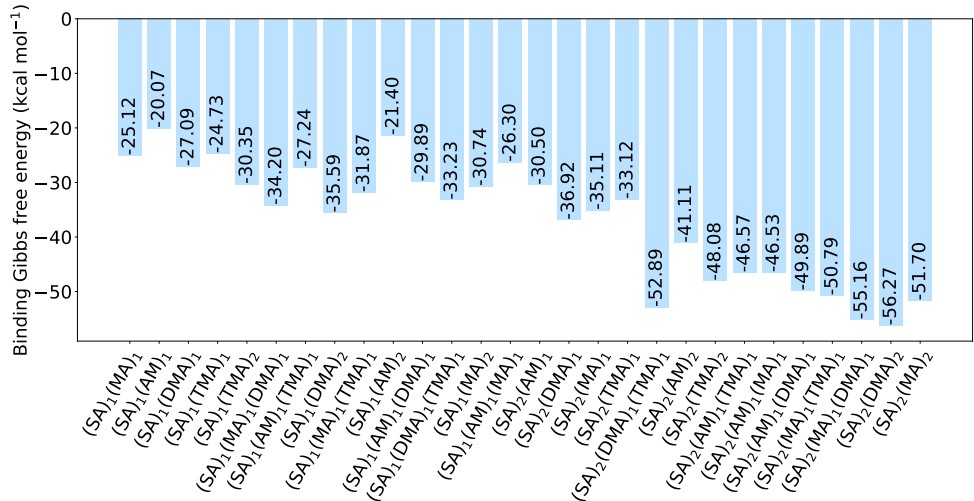

**Figure 7.** The binding free energy of the $(SA)_{1-2}(Base)_1(CA)_3$ clusters, calculated at the DLPNO-CCSD(T$_0$)/aug-cc-pVTZ//$\omega$B97X-D/6-31++G(d,p) level of theory with quasi-harmonic cutoff of 100 cm$^{-1}$, at 298.15 K and 1 atm. Carboxylic acid has been omitted in the labels for readability.

Logically, the inclusion of 3 carboxylic acid groups has a stabilizing effect, just as the inclusion of more SA and base does. Following the trend from earlier of DMA producing the most stable clusters, $(SA)_2(DMA)_2(CA)_3$ is the most stable cluster with $\Delta G = -56.27$ kcal mol$^{-1}$. Furthermore, DMA is found as a component in the three clusters that are lowest
in free energy, with $(SA)_2(MA)_1(DMA)_1(CA)_3$ having $\Delta G = -55.16$ kcal mol$^{-1}$ and $(SA)_2(DMA)_1(TMA)_1(CA)_3$ having $\Delta G = -52.89$ kcal mol$^{-1}$. AM, on the other hand, is again the base that produces the clusters with the highest free energy, both alone and in combination with other bases. The two clusters highest in free energy are $(SA)_1(AM)_2(CA)_3$ with $\Delta G = -21.40$ kcal mol$^{-1}$ and $(SA)_1(AM)_1(CA)_3$ with $\Delta G = -20.07$ kcal mol$^{-1}$. Two SA molecules lead to more stable clusters than two DMA molecules, as $(SA)_2(DMA)_1(CA)_3$ has a slightly lower free energy compared to $(SA)_1(DMA)_2(CA)_3$ with a
$\Delta G$ of -36.92 and -35.59 kcal mol$^{-1}$, respectively. While this is consistent with previous studies (Olenius et al., 2013; Elm, 2017), we see that the inclusion of the organic acids, decreases the difference between the two paths. The clusters lowest in free energy all have two SA molecules and two bases, as the inclusion of more SA and base gives rise to more acid-base interactions, as is illustrated in Figure 8.

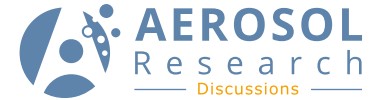

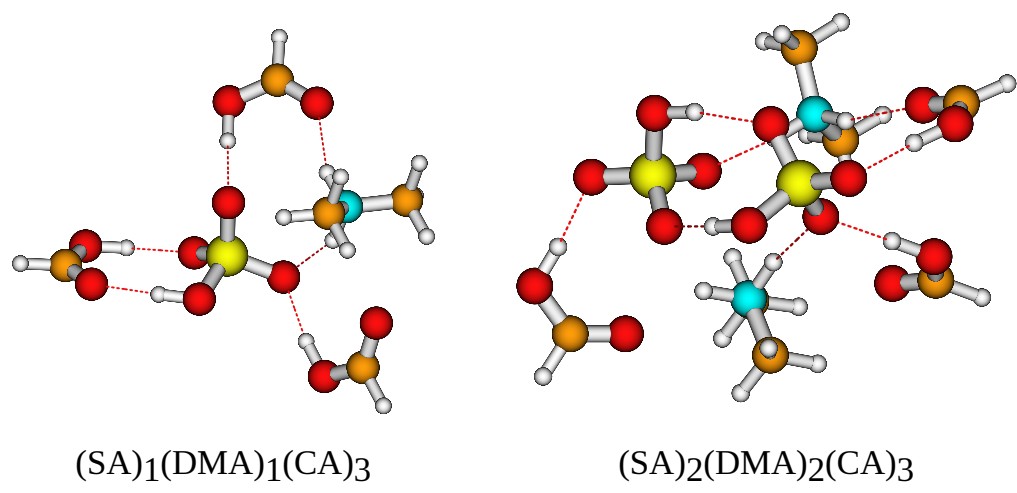

$(SA)_1(DMA)_1(CA)_3$ $\qquad$ $(SA)_2(DMA)_2(CA)_3$

**Figure 8.** The $(SA)_{1-2}(DMA)_{1-2}(CA)_3$ cluster geometries lowest in free energy. Calculated at the DLPNO-CCSD($T_0$)/aug-cc-pVTZ//$\omega$B97X-D/6-31++G(d,p) level of theory with quasi-harmonic cutoff of $100$ cm$^{-1}$, at 298.15 K and 1 atm. White = hydrogen, brown = carbon, red = oxygen, yellow = sulfur, blue = nitrogen. CA is short for Carboxylic Acid.

While the clusters become lower in free energy as they increase in size, it is not a guarantee that they are stable against evaporation in the atmosphere. For instance, the subsequent addition of more carboxylic acid groups to a the $(SA)_1(DMA)_1$ cluster leads to a decrease of $-8.04$, $-4.54$ and $-3.13$ kcal mol$^{-1}$ for the first, second and third addition, respectively. Hence, if these are treated as individual molecules it is unlikely that they form stable clusters at realistic atmospheric conditions. However, if all three functional groups are treated as single OOM, the free energy for adding a hypothetical idealized tricarboxylic to

the $(SA)_1(DMA)_1$ cluster is $\Delta G = -15.71$ kcal mol$^{-1}$. This is a very strong binding and would correspond to a quite stable cluster. Hence, by employing the "cluster-of-functional-groups" approach we have identified that tricarboxylic acids are likely candidates for forming stable clusters with SA and bases. This conclusion was already hypothesized, but not explicitly proven in our earlier work (Elm et al., 2017a).

### 3.5 Cluster formation potentials

Based on the calculated thermochemistry in the previous sections we can study how a hypothetical OOM composed of three carboxyl groups can potentially stablize sulfuric acid – base clusters. Letting the three carboxyl groups represent a single OOM we simulated the cluster formation potential ($J_{\text{potential}}$) of the $(SA)_{1-2}(base)_{1-2}(OOM)_1$ systems, with base = AM, MA, DMA and TMA. To allow direct comparability, we conformed with our previous studies and used the same vapour concentrations as in the clusteromics series of papers (Elm, 2021a, b, 2022; Knattrup and Elm, 2022; Ayoubi et al., 2023). Sulfuric acid was

fixed at $10^6$ molecules cm$^{-3}$ and the concentration of the bases was studied giving two extremes with a "lower limit" and an "upper limit". These were set as follows: AM (10 ppt, 10 ppb), MA (1 ppt, 100 ppt) and DMA/TMA (1 ppt, 10 ppt). It should be noted that the low concentration limit likely best represents the actual concentrations observed in the ambient atmosphere.





The simulations were performed at 278.15 K and 1 atm using the settings described in Section 2.4. Figure 9 presents the simulated cluster formation potentials ($J_{potential}$) for the four bases as a function of OOM concentration (0-10 ppt).

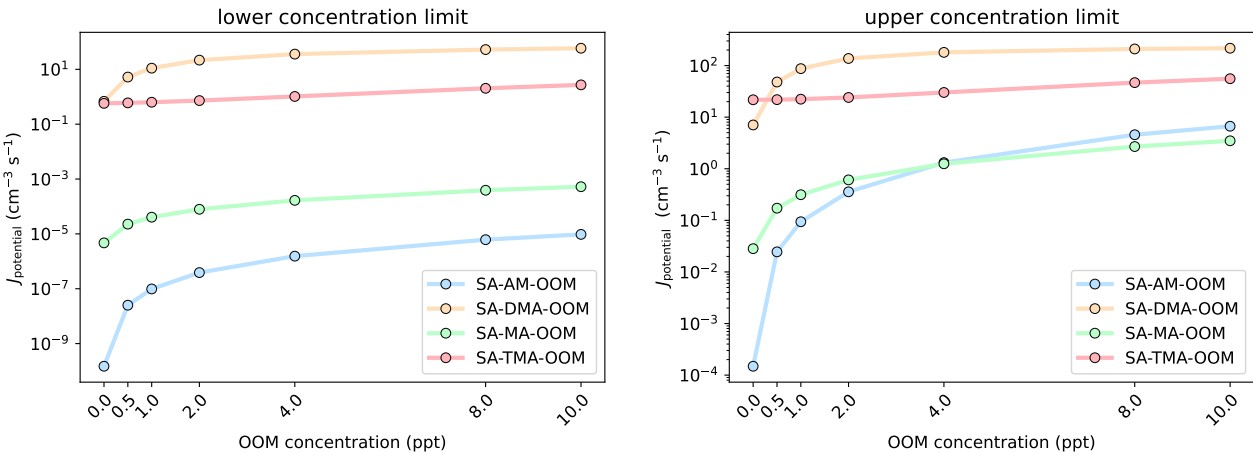

**Figure 9.** Simulated cluster formation potentials (in clusters $cm^{-3}$ $s^{-1}$) as a function of oxygenated organic molecule mixing ratio, in the lower concentration limit (*left*) and upper concentration limit (*right*). The simulations are performed at 278.15 K and 1 atm.

It is seen that the influence of the added OOM is highly dependent on the base in the SA–base–OOM systems. In the lower concentration limit the $J_{potential}$-value increases around one order of magnitude with 10 ppt OOM present for MA and DMA. This is roughly four orders of magnitude for the SA–AM–OOM systems. However, the increase is from $1.49 \times 10^{-10}$ to $9.59 \times 10^{-6}$ $cm^{-3}$ $s^{-1}$, implying that the absolute cluster formation potential is still negligible. The cluster formation potential of the SA–TMA–OOM system is found to be more or less unaffected by the presence of the OOM and thereby does not increase much with increased OOM concentration. This is caused by the three bulky methyl groups in TMA, which impede the attachment of the OOM due to steric hindrance (See SI). The formation potentials present the following trend at 10 ppt of OOM: DMA > TMA > MA > AM. In the upper concentration limit we see very similar trends, except that the SA–AM–OOM system begins to overtake the SA–MA–OOM system, due to the very high mixing ratio of 10 ppb of AM.

Inspecting the fluxes out of the systems, in the low concentration regime, show that the OOM (at 1 ppt) contribute to 97.96, 98.86, 72.44 and 7.52 % of the outgrowing fluxes for the AM, MA, DMA and TMA systems, respectively. The fluxes for each system at various concentrations can be seen in the SI. Hence, the more weakly bound the SA–base system is without thes OOM present, the more the OOM will contribute to the cluster formation potential. At 10 ppt of OOM 100 % of the outgrowing clusters contain an OOM for AM, MA and DMA, both in the lower and upper concentration regime of the bases. At 10 ppt of OOM the outgrowing clusters containing an OOM for TMA is 78.97 and 87.22 % in the lower and upper concentration limits. Again, it should be noted that the simulated $J_{potential}$-values show the potential to grow to larger sizes and not the actual nucleation rate. Hence, whether the OOM will evaporate from larger clusters or further contribtute to the cluster growth remains unknown.





Overall, we see that the hypothetical OOM substantially contributes to the cluster formation potential, but as this is an idealized compound, explicit SA–base–tricarboxylic acid clusters should be further studied. Cluster formation with 3-methyl-1,2,3-butanecarboxylic acid (MBTCA) (Müller et al., 2012) has previously been carried out for SA–MBTCA clusters (Ortega et al., 2016; Elm et al., 2017b) and SA–AM–MBTCA clusters (Myllys et al., 2017), but it remains ambiguous whether MBTCA is important for cluster formation. Recent evidence suggested that SA–MBTCA clusters primarily grow along the MBTCA coordinate, i.e. without the participation of SA (Elm, 2019b). Other potential atmospheric relevant tricarboxylic acids could be the 3-carboxyheptanoic acid (CHA) compound identified from limonene oxidation (Jaoui et al., 2006; Yasmeen et al., 2011). Recent work indicate that the CHA might be involved in SA–CHA cluster formation (Tan et al., 2022) and it would be worth to explicitly study SA–base–CHA clusters further. Finally, there has been identified a large accretion product, known as pinyl diaterpenylic ester (PDPE), from $\alpha$-pinene oxidation (Kristensen et al., 2013). Such a large compound has newer been studied in atmospheric cluster formation and would be worth investigating in the future. This is also aligned with experimental observations, that deem large accretion products as the most likely OOMs to drive nucleation (Lehtipalo et al., 2018; Dada et al., 2023).

## 4   Conclusions

Using quantum chemical calculations we have employed a new method, denoted the "cluster-of-functional-groups" approach, for studying the role of oxygenated organic molecules (OOMs) in organic enhanced atmospheric cluster formation. Studying all combinations of organic donor/acceptor functional groups and their intermolecular interactions with sulfuric acid – base clusters it is found that carboyxyl groups leads to the most stable clusters. Based on our findings, we study the cluster formation potential of a hypothetical tricarboxylic acid molecule, represented as a cluster of three carboxyl groups. We find that the proxy OOM composed of three carboxylic acid groups are present almost in all the relevant outgrowing clusters. Thereby the OOM is directly participating in SA–base-OOM cluster formation. Hence, the role of explicit tricarboxylic acids should further studied, and especially large flexible tricarboxylic acid accretion products might be important for cluster formation.

We hypothesize that a "cluster-of-functionals-groups" composed of multiple carbonyls, alcohols and hydroperoxide groups might rival the binding strength of three carboxylic acids in SA–base clustering and should be further studied in the future. The approach should also be extended to study ion-induced nucleation, which might subtantially enhance pure organic nucleation. In addition, the effect of temperature should be investigated, as all the functional groups will bind more strongly at lower temperatures.

Overall, our presented "cluster-of-functional-groups" approach is general and applicable to other topics where multifunctional compounds are important and where the exact chemical structure remains unknown. Hence, we speculate that the work could also be extended to study the potential molecular structures that can enhance ice nucleation in the form of identifying potential functional groups that facilitate the formation of hexagonal ice $I_h$ structures.

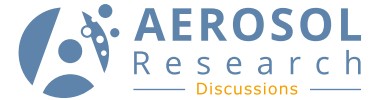

*Data availability.* All the calculated structures and thermochemistry are available in the Atmospheric Cluster Database (ACDB) at: https://github.com/elmjonas/ACDB/tree/master/Articles/pedersen24_org_interactions

*Author contributions.* Conceptualization: J.E.;

Methodology: A.N.P, Y.K., J.E.;

Formal analysis: A.N.P, Y.K.;

Investigation: A.N.P, Y.K.;

Resources: J.E.;

Writing - original draft: A.N.P, J.E.;

Writing - review & editing: A.N.P, Y.K., J.E.;

Visualization: A.N.P, Y.K.;

Project administration: J.E.;

Funding acquisition: J.E;

Supervision: J.E.

*Competing interests.* J.E is a member of the Editorial Board of Aerosol Research. The remaining authors have no conflict of interests to declare.

*Acknowledgements.* The authors thank the Independent Research Fund Denmark grant number 9064-00001B for financial support. This work was funded by the Danish National Research Foundation (DNRF172) through the Center of Excellence for Chemistry of Clouds.

The numerical results presented in this work were obtained at the Centre for Scientific Computing, Aarhus https://phys.au.dk/forskning/faciliteter/cscaa/.

The authors thank Professor Merete Bilde for insightful discussions of the present work.





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
