# Peer review of "Supporting Information: A Cluster-of-Functional-Groups Approach for Studying Organic Enhanced Atmospheric Cluster Formation"

_Aerosol Research, 2024_

## Author Comment (AC1)

**Response to Reviews**

We highly appreciate the positive comments from both the reviewers and all the points have been addressed in the revised paper. We hope that the following responses are satisfying and that the paper can be accepted for publication. The reviewers' comments have been reproduced in blue text below, followed by our point-by-point replies.

**RC1: 'a small suggested improvement to the model', Theo Kurtén, 04 Mar 2024**

First, a caveat. As I collaborate quite a lot with professor Elm, this comment should really be considered a "community comment" rather than a reviewer comment. I.e. the authors should feel free to disregard my comment, and the editors should place a low weight on my review (either way) when deciding on what to do with the paper.

This is a nice paper introducing a useful new concept, but I have a suggestion on how to take it further (either in this manuscript, or in future work). Specifically, it might be useful to discuss and treat the proposed additivity in terms of the two components of the binding/-formation free energy: the enthalpy and the entropy.

In the absence of steric constraints (e.g. all three groups of a tricarboxylic acid being prevented from binding optimally, compared to three separate HCOOH), and ignoring intermolecular H-binding in the reactant organics, the binding energy should indeed be roughly additive (and thus also the enthalpy, apart from the rather minor pV-term, which I'd like to thank Lauri Franzon in my group for pointing out). So far so good.

However for the clustering entropy, the major contribution comes from the loss of translational and rotational degrees of freedom (and their conversion into lower-entropy vibrational degrees of freedom). Three of each (i.e. six in total) are lost per clustering molecule, so e.g. for 3 x HCOOH the total loss is 18 high-entropy degrees of freedom, while for a tricarboxylic only 6 would be lost. The entropy loss upon clustering is thus very probably NOT additive, or at least it could/should be split into two components:

-a definitely non-additive term coming from the above-mentioned loss (this should be counted only once per condensing molecule)

-another, possibly additive, term originating from some of the flexible internal rotations (especially important in large OOMs), becoming more constrained during clustering.

**Author reply:**
We completely agree with Theo Kurtén in this aspect and are indeed well-aware that the entropy is not necessarily additive. We are currently working on calculating clusters containing larger realistic tricarboxylic acid molecules (1,2,3-butanecarboxylic acid, 3-carboxyheptanoic

acid and pinyl diaterpenylic ester). With these systems we can explicitly quantify the change in entropy by removing the backbone of tricarboxylic acid molecules. As the manuscript already contain a quite substantial amount calculations, and this is ongoing work, we would prefer to report these systems in a separate manuscript. However, we will address the potential non-additivity of the entropy contribution in a more pragmatic manner as suggested by Theo in the comments below.

As the translational entropy loss is determined directly from the molecular mass, and the rotational entropies can be estimated from very crude (e.g. molecular mechanics - level) simulations, or alternatively fitted to the datasets the authors already have, perhaps some parametrisation of the clustering entropy along the lines proposed above could be envisioned, in order to improve the model? Having said that, the current admirably simple model probably benefits from some degree of cancellation of error. As discussed above, the entropy penalty of clustering one tricarboxylic (with say a SA-DMA "core") is considerably smaller than that of clustering three HCOOH. So based on that, the delta-G for adding the tricarboxylic might be much more negative than the -15 kcal/mol value quoted here. At the same time, it's unlikely that all three carboxylic acids groups of any real tricarboxylic can simultaneously reach the ideal bonding geometries shown in e.g. figure 8. So also the enthalpy gain will be less than in the "perfect additivity" assumption. Furthermore, the model (if I understand it correctly) completely neglects possible intramolecular H-bonds inside the reactant OOM, which tend to decrease the favourability of their clustering reactions (as some of the H-bonding capacity is already used up, so to speak). Almost certainly any real tricarboxylic will tend to have at least some interactions between some of the groups already in the organic monomer.

**Author reply:**
We completely agree with these aspects and believe that it would be worth mentioning these deficiencies (intramolecular hydrogen bonds and geometric constraint), as well as the non-additivity of entropy contribution in the model. We have modified the following paragraph on page 4.

**Changed paragraph, page 4:**

**From:**
However, we note the caveat that the "cluster-of-functional-groups" approach assumes that the binding free energies of the individual groups are additive. This might not necessarily be the case for realistic atmospheric OOMs, but the approach can still be employed for screening purposes and to yield some indication of which combinations of functional group might potentially be important in atmospheric cluster formation.

**To:**
However, we note the caveat that the "cluster-of-functional-groups" approach assumes that the binding free energies of the individual groups are additive. This might not necessarily be the case for realistic atmospheric OOMs, where several effects potentially make the free energies deviate from additivity and there can be expected some degree of cancellation of errors. For instance, the enthalpy contribution is expected to be more or less additive given that there are no intramolecular hydrogen bonds in the OOM. In addition, it is unlikely that the multiple moieties of any realistic OOM can simultaneously reach the ideal contact points

without introducing some strain in the backbone, which will also lead to a higher enthalpy. The entropy contribution will most likely not be additive as a major contribution comes from the loss of translational and rotational degrees of freedom i.e. the conversion into lower-entropy vibrational degrees of freedom. For each clustering functional group six high-entropy degrees of freedom are lost, while only a total of six high-entropy degrees of freedom are lost in the OOM. Despite these deficiencies, the "cluster-of-functional-groups" approach can still be employed for screening purposes and to yield some indication of which combinations of functional groups might potentially be important in atmospheric cluster formation.

This leads me to a suggestion (that I only realised after writing the above text): perhaps most or all of this (with the possible exception of more complex steric effects in huge OOMs) could be modelled by using e.g. $(HCOOH)_2$ or $(HCOOH)_3$ (or the corresponding clusters of other model organics) as the model reactants? I.e. take the enthalpy gain from a comparison of $X + (HCOOH)_n => X(HCOOH)_n$, and then split the entropy as suggested above, counting the "loss-of-translation-and-rotation" penalty only once? (X stands for the inorganic "core" here, e.g. SA*DMA). Or - even simpler and better - the second entropy term, i.e. the constraining of flexible internal rotations, would actually already be partly taken into account in the $(HCOOH)_2$ and $(HCOOH)_3$ clusters... so actually just straightforwardly computing the delta-G for the reaction $X + (HCOOH)_n => X(HCOOH)_n$ would provide a decent proxy for the delta-G of the addition of a tricaboxylic without steric constraints, BUT making perfect internal H-bonds. (Thus providing some cancellation of errors also for this somewhat more nuanced additivity approach.) As the authors should have most of the data already (they certainly have at least $(HCOOH)_2$, and $(HCOOH)_3$ should be easy enough to generate), maybe quickly check what type of numbers this approach gives (at least for the carboxylics), and compare with what they have in the present paper?

**Author reply:**
This sounds like a very pragmatic approach to model the loss of high-entropy degrees of freedom. As suggested, we have tested how using the cluster of three carboxylic acid groups as the reactant instead of three individual groups influence the binding free energies. I.e we look at the following reactions:

$$X + n \times HCOOH \longrightarrow (X)(HCOOH)_n \tag{1}$$
$$X + (HCOOH)_n \longrightarrow (X)(HCOOH)_n \tag{2}$$

Here reaction (1) is the usual way of calculating the binding free energies. Looking at the binding free energy difference between reaction (1) and (2) we get:

$$\Delta\Delta G = \Delta G\Big[(HCOOH)_n\Big] - \Delta G\Big[n \times HCOOH\Big] \tag{3}$$

Surprisingly, computing this value at 298.15 K and 1 atm leads to a perfect cancellation of errors with a $\Delta\Delta G$-value of -0.06 kcal/mol. However, as the enthalpy and entropy terms are different for reaction (1) and (2), we can expect a different temperature dependence. Recalculating $\Delta\Delta G$ at 278.15 K and 1 atm leads to +1.5 kcal/mol difference. To comment on this aspect we have added the following paragraph on page 12:

**Added paragraph, page 12:**

It should be noted that this approach neglects the loss of high-entropy translation-and-rotation degrees of freedom (see discussion in Section 2.3). A pragmatic approach to remedy this effect would be to model the single OOM as a cluster of the three carboxylic acid groups. Essentially, we want to compare the following two reactions:

$$(SA)_1(DMA)_1 + 3 \times HCOOH \longrightarrow (SA)_1(DMA)_1(HCOOH)_3 \qquad (3)$$

$$(SA)_1(DMA)_1 + (HCOOH)_3 \longrightarrow (SA)_1(DMA)_1(HCOOH)_3 \qquad (4)$$

Calculating the binding free energy difference between reaction (3) and (4) we obtain:

$$\Delta\Delta G = \Delta G\Big[(HCOOH)_3\Big] - \Delta G\Big[3 \times HCOOH\Big] \qquad (4)$$

Computing this value at 298.15 K and 1 atm leads to a perfect cancellation of errors with a $\Delta\Delta G$-value of -0.06 kcal mol$^{-1}$. However, as the enthalpy and entropy terms are different for reaction (3) and (4), we can expect a different temperature dependence. Recalculating $\Delta\Delta G$ at 278.15 K and 1 atm leads to +1.5 kcal mol$^{-1}$ difference. Hence, there is very little difference between the two methods of calculating the binding free energies and we will in the following stick to the simple method given by reaction (3). Nevertheless, ...

**RC2: 'Comment on ar-2024-6', Anonymous Referee #2, 10 Mar 2024**

Elm et al. utilize a novel approach called the "cluster-of-functional-groups" to investigate the involvement of oxygenated organic molecules (OOMs) in atmospheric cluster formation. By examining these interactions, they aim to identify the structural characteristics of OOMs contributing to particle formation. Their study shows that clusters with multiple carboxyl groups are notably stable. Through simulations, they explore how OOMs with multiple carboxyl groups can stabilize sulfuric acid - base clusters, suggesting potential tracer compounds for future research. The presented "cluster-of-functional-groups" approach is novel and applicable in the study of atmospheric aerosols. The most part of this manuscript is well written and of broad interest to the readership of Aerosol Research. I recommend publication in Aerosol Research after the following comments have been addressed.

Specific Comments:

Comment 1: Page 4 lines 91-92: "For each protonation state in the cluster, 1000 local minima were saved." The meaning of "each protonation state in the cluster" may be confused. Did the author investigate all possibilities of acid-base reactions within the same cluster? Please provide more detailed information on this aspect.

**Author reply:**
We agree with the reviewer that this is perhaps ambiguously written. We sampled the clusters using both neutral, anionic, and cationic monomers. In all cases the overall cluster charge was kept neutral. To clarify this aspect we have modified the sentence.

**Modified Sentence, page 4:**

**From:**
For each protonation state in the cluster, 1000 local minima were saved.

**To:**
We employed both neutral, anionic, and cationic monomers in all combinations leading to overall neutral clusters and saved 1000 local minima for each combination.

Comment 2: Page 4 lines 100-103: "Small organic molecules were chosen to represent the functional groups that act as hydrogen bond donors (alcohol ($CH_3OH$) and peroxide ($CH_3OOH$)), as well as functional groups that act as hydrogen bond acceptors (ether ($CH_3OCH_3$), epoxide ($C_2H_4O$), aldehyde ($CH_3CHO$), ketone ($CH_3COCH_3$), acid anhydride ($CH_3C(=O)OC(=O)CH_3$) and ester ($COOCH_3$). Besides these groups, carboxylic acid, ($HCOOH$), which is both an acceptor and a donor, was also included." This method is innovative, however, I still have a concern. Earlier, the authors mentioned that "not a single OOM has definitively been proven to participate in nucleation in the planetary boundary layer (Elm et al, 2023). The lack of progress could be ascribed to the fact that previous work have been looking at the wrong compounds. All studies have been performed on the organic monomers, while recent evidence from the CLOUD chamber has shown that it is in fact the covalently bound organic dimers". This implies that monomers and dimers with similar functional groups may exhibit completely different nucleation capabilities. Can the "cluster-of-functional-groups" method

adequately consider the difference between monomers and dimers? I hope the authors can provide a brief discussion on this concern.

**Author reply:**
We appreciate the comment. While the "cluster-of-functional-groups" method *per se* cannot directly distinguish between monomers and dimers it can give a hint to the identity of compound. For instance, if the functional groups are positioned far from each other in space to yield a stable cluster, it is very unlikely that a monomer with a short carbon backbone would be able to connect the functional groups. To elaborate on this aspect we have modified the following sentence on page 4.

**Modified Sentence, page 4:**

**From:**
The strength of this approach is that we do not need to explicitly consider the origin of the OOM i.e anthropogenic (aromatics) or biogenic (isoprene/terpenes) as the approach will inherently identify the structural patterns that are important directly based on their ability to participate in cluster formation.

**To:**
The strength of this approach is that we do not need to explicitly consider the origin of the OOM i.e anthropogenic (aromatics) or biogenic (isoprene/terpenes) or whether it is a monomer or an accretion product as the approach will inherently identify the structural patterns that are important directly based on their ability to participate in cluster formation. However, if the functional groups are optimally oriented far from each other, it could allude to that the potential compound is an accretion product.

Comment 3: Page 5 lines 127-130: "For our systems, we chose the clusters with one additional acid compared to the cluster sizes we have data for $[(SA)_3(base)_2, (SA)_2(OOM)_2(base)_2$ and $(SA)_3(OOM)_1(base)_2]$. These outgrowing cluster sizes are quite small and therefore artificially stabilize the systems as the critical cluster size is not necessarily captured well." Why were different sizes of outgrowing clusters used for the SA-base and SA-base-OOM system? For the ACDC simulations, setting different sizes of outgrowing clusters may directly affect the simulated cluster formation rates of SA-base and SA-base-OOM system, thereby overestimating or underestimating the impact of OOM. Please provide more information about these settings.

**Author reply:**
We sat the boundary conditions to clusters with one more acid than the clusters we have data for. This is a quite usual procedure and has been applied throughout the clusteromics series of papers as well. The reason for this choice is that clusters with more bases than acids are usually unstable and would lead to a high overestimation of the $J_{\text{potential}}$ rates if included. In addition, in order to actually see the effect of the OOM we had to "constrain" the possibility of having one OOM attached at the outgrowing cluster. This implies that clusters with an OOM must leave the system either by collision with another OOM or SA. Consequently, this means that the growth path involving OOMs needs one more molecule to grow out of the system and thereby the role of OOMs shown in Figure 9 is likely slightly underestimated. To further elaborate on these aspects we have added the following clarifying sentence.

**Added Sentence, page 5:**
We exclude outgrowing collisions with bases as clusters composed of more bases than acids are usually not stable. The outgrowing clusters containing OOMs have one more molecule present compared to the pathway without OOM. This will slightly underestimate the contribution of OOM to the relative simulated $J_{\text{potential}}$ rates.

Comment 4: Page 12 lines 222-227: "Hence, if these are treated as individual molecules it is unlikely that they form stable clusters at realistic atmospheric conditions. However, if all three functional groups are treated as single OOM, the free energy for adding a hypothetical idealized tricarboxylic to the $(SA)_1(DMA)_1$ cluster is $\Delta G = -15.71$ kcal mol$^{-1}$. This is a very strong binding and would correspond to a quite stable cluster. Hence, by employing the "cluster-of-functional-groups" approach we have identified that tricarboxylic acids are likely candidates for forming stable clusters with SA and bases." I have doubts about the hypothesis that the binding free energies of the individual functional groups are additive:

1. As shown in Figure 8, the introduction of three CAs into SA-DMA can saturate the cluster. However, these three CAs are positioned on the outer side of SA-DMA in three different directions, implying that a single tricarboxylic acid may have difficulty achieving the same effect. Would such a possibility affect the author's predictions?

**Author reply:**
We agree with the reviewer that the binding free energies are most likely not additive. We also agree that a realistic OOM with three carboxylic acid groups (even a large accretion product) will most likely not be able to reach these positions without introducing some strain in the carbon backbone. We elaborated on this aspect in the comment above from Theo Kurtén as well. See the modified paragraph on page 4 below:

**Modified paragraph, page 4:**
However, we note the caveat that the "cluster-of-functional-groups" approach assumes that the binding free energies of the individual groups are additive. This might not necessarily be the case for realistic atmospheric OOMs, where several effects potentially make the free energies deviate from additivity and there can be expected some degree of cancellation of errors. For instance, the enthalpy contribution is expected to be more or less additive given that there are no intramolecular hydrogen bonds in the OOM. In addition, it is unlikely that the multiple moieties of any realistic OOM can simultaneously reach the ideal contact points without introducing some strain in the backbone, which will also lead to a higher enthalpy. The entropy contribution will most likely not be additive as a major contribution comes from the loss of translational and rotational degrees of freedom i.e. the conversion into lower- entropy vibrational degrees of freedom. For each clustering functional group six high-entropy degrees of freedom are lost, while only a total of six high-entropy degrees of freedom are lost in the OOM. Despite these deficiencies, the "cluster-of-functional-groups" approach can still be employed for screening purposes and to yield some indication of which combinations of functional group might potentially be important in atmospheric cluster formation.

2. Compared to three CAs, a tricarboxylic acid is likely to introduce some parts with minimal contributions to the nucleation process. Will these parts inhibit nucleation?

**Author reply:**
This is a tricky question and without explicit data on realistic tricarboxylic acids it is impossible to quantify and would be pure speculation on our part. While a long carbon backbone might introduce hydrophobic parts that will repel other key nucleation precursors it also leads to a favourable entropy loss compared to the three individual carboxylic acid. As shown above in the modified paragraph on page 4 there are several contributions that go in opposite directions depending on the exact given compound. We are currently working on calculating clusters containing larger realistic tricarboxylic acid molecules (1,2,3-butanecarboxylic acid, 3-carboxyheptanoic acid and pinyl diaterpenylic ester). At the moment we would prefer not to comment on this in the current manuscript, as it would just be speculation. However, with these new explicit systems we will be able to quantify this aspect in the future.

3. The author's hypothesis and conclusion demonstrate great innovation and foresight. If feasible, I suggest supplementing some data of the binding free energies of SA-DMA-tricarboxylic acid clusters to validate this conclusion.

**Author reply:**
We highly appreciate the kind comment. Calculations with explicit tricarboxylic acids are quite heavy and as the current paper already includes an extensive amount of data we would prefer not to include additional new calculations here. However, we are currently working on this aspect, but would prefer to include that as a separate coherent story, once finished.

Comment 5: Page 12 lines 231-233: "Letting the three carboxyl groups represent a single OOM we simulated the cluster formation potential ($J_{potential}$) of the $(SA)_{1-2}(base)_{1-2}(OOM)_1$ systems, with base = AM, MA, DMA and TMA." To let the three carboxyl groups represent a single OOM, did the author treat the binding free energy of OOM as the sum of three carboxylic acid groups? If so, then I would like to know how parameters such as the radius and mass of the OOM are set during ACDC simulations, as these directly affect the simulation results.

**Author reply:**
We agree that this aspect should have been added to the manuscript. Indeed, an OOM will have a different radius and hence a different collision rate coefficient in the ACDC simulations. The radius is calculated from the liquid density and mass of the system and these are of course unknown for the unknown OOM. Hence, as an approximation we assumed that the liquid density was the same as for formic acid and set the mass to three times that of formic acid. We have added this clarification to page 13.

**Added sentence, page 5:**
We calculated the collision coefficient of the OOM in the ACDC simulations based on the liquid density for formic acid and three times the mass of formic acid.

Minor Comments:

Comment 1: Page 2 line 37: "The puzzle of the role of organics in aerosol nucleation originate from the fact that there exist a myriad of OOMs in the atmosphere." "exist" → "exists".

Comment 2: Page 2 line 50: "The lack of progress could be ascribed to the fact that previous work have been looking at the wrong compounds". "have"→"has".

Comment 3: Page 3 line 83: "The simulations were done at 278.15 K with a constant coagulation sink of - 1.6 × 10-3 s-1". The format of the value "1.6 × 10-3 s-1" is inconsistent with the format of the main text. Similar problems occur repeatedly in the main text. Please correct it.

Comment 4: Page 5 line 131: "These rates illustrates the potential of the cluster to grow to larger sizes and corresponds to an upper-bound on the formation rate." "illustrates"→"illustrate".

Comment 5: Page 6 line 142: "However, this value correspond to a high evaporation rate of the formic acid dimer" "correspond"→"corresponds".

Comment 6: Page 7 line 153: "... where there is two carboxylic acid groups in total as this fully saturates the SA molecule." "is"→"are".

Comment 7: Page 14 line 268: "Such a large compound has newer been studied in atmospheric cluster formation and would be worth investigating in the future." "newer"→"never".

**Author reply:**
All these issues have been corrected in the revised manuscript.